# A Note on Singularity Avoidance in Fourth-Order Gravity

**Luca Fabbri** 

DIME, Sez. Metodi e Modelli Matematici, Università di Genova, Via all'Opera Pia 15, 16145 Genova, Italy; fabbri@dime.unige.it

**Abstract:** We consider the fourth-order differential theory of gravitation to treat the problem of singularity avoidance: studying the short-distance behaviour in the case of black-holes and the big-bang we are going to see a way to attack the issue from a general perspective.

**Keywords:** higher-order derivative torsion-gravity; singularity

## 1. Introduction

With the detection of gravitational waves, each original experimental prediction of Einstein gravity has now been settled. In fact, if dark matter is indeed a form of matter, and not a gravitational effect, there are no observational open issues left in modern gravity. Nevertheless, from a purely theoretical perspective, there are yet two problems that need fixing, and which are, more or less, connected: one is solving the nature of singularity formation, which seems to be an occurrence that is unavoidable, in light of the Hawking-Penrose theorem; the other, to have gravity made into a renormalizable theory, hence to fit, with the standard model of particles, into one single framework.

Because renormalizability needs all fields with a kinetic term of mass dimension 4 and because for gravitation the mass dimension is assigned to be equal to 2 for curvature terms, Einsteinian gravitation cannot meet the necessary requirements. However, the solution is simple: raise to 2 the number of curvatures appearing in the Lagrangian.

This strategy has led to a variety of extensions of Einstein gravity, of which the $f(R)$ types are just the most famous to have arisen in recent times (for a general overview, we refer the reader to [1–5] and references therein).

This type of gravitational extension may meet also the criteria to solve the problem of singularity formation [6].

However, by not stopping at two curvatures, they tend to include in the Lagrangian terms that have too high a differential order for renormalizability. Renormalizability is instead granted for theories displaying a conformal invariance [7–9]. Yet, when torsion is included, it is not difficult to see that this theory is not continuous, in the sense that conformal gravity with torsion taken in its torsionless limit does not give the conformal gravity that we would have without having torsion in the first place [10].

This problem is in fact very general, affecting virtually everyone of the higher-order theories of gravitation [11].

As a matter of fact, a thorough application of the principle requiring continuity for the torsionless limit of any torsion gravity imposes severe restrictions to the possible forms that the gravitational Lagrangian can have [12].

Remarkably, by following this principle the Lagrangian found in [12] is the one that grants at most 2 curvatures and the renormalizable kinetic term for torsion [13,14].

So a question naturally arises, asking whether this Lagrangian can also provide some solution to the singularity problem. The answer is partly positive, because a renormalizable torsion theory, where torsion is an axial-vector field coupled to the spin of spinors, ensures the conditions for which the Hawking-Penrose inequality is violated and singularities no longer unavoidable [15] (and as a matter of fact, even the Higgs field can do the same [16]),

so long as black holes are considered. Singularities that involve the big bang instead are more delicate because symmetry arguments may be used to demonstrate that torsion does not impact the energy density for the gravitational field equations [17] (for the Higgs the argument is that at the energy scales of the big bang no spontaneous symmetry breaking has occurred yet). Thus the question, could we have a way to avoid singularity formation, or at least its inevitability, even for early cosmological scenarios?

## 2. Fourth-Order Differential Torsion-Gravity

We start by recalling the results of [12]. In this work it was shown we get continuity, thus the torsionless limit of torsion gravity gives pure gravity, only if the Lagrangian is restricted to a very special type: this Lagrangian is

$$\mathscr{L} = -\tfrac{1}{4}(\nabla_\alpha W_\nu - \nabla_\nu W_\alpha)(\nabla^\alpha W^\nu - \nabla^\nu W^\alpha) - YR_{\alpha\mu}R^{\alpha\mu} - ZR^2 + \tfrac{1}{2}M^2 W_\nu W^\nu - R + \\ + i\overline{\psi}\gamma^\mu\boldsymbol{\nabla}_\mu\psi - X\overline{\psi}\gamma^\mu\boldsymbol{\pi}\psi W_\mu - m\overline{\psi}\psi \tag{1}$$

where $W_\alpha$ is the axial-vector field known as torsion, $R_{\alpha\mu}$ and $R$ are the Ricci tensor and scalar, $\psi$ and $\overline{\psi}$ the pair of adjoint spinor fields. The metric is given by $g_{\mu\nu}$ and the connection $\Lambda^\rho_{\mu\nu}$ is used to compute the covariant derivatives, with $\gamma^\mu$ being the Clifford matrices ($\boldsymbol{\pi}$ is normally denoted by a gamma with an index 5, but because in the space-time this index has no meaning we will employ the definition with no index). Finally, $m$ and $M$ are the mass of the spinor and torsion with $X$ being the torsion-spinor coupling constant. Notice that the case $Y = Z = 0$ reduces the Lagrangian to its least-order derivative form, but in general this Lagrangian has 2 curvatures and so it has fourth-order differential character in the metric. Remark also that the term $R$ should be multiplied by a constant with mass dimension 2 and that is the Newton constant, although we have normalized it to unity and therefore it does not explicitly show within the Lagrangian. Because the metric is dimensionless, every curvature counts for a mass dimension 2 so that the general kinetic term is mass dimension 4 with strong consequences for the requirement of renormalizability as discussed in references [13,14].

Varying the torsion axial-vector, the metric tensor and the spinor field, we get respectively the field equations

$$\nabla_\rho(\partial W)^{\rho\mu} + M^2 W^\mu = X\overline{\psi}\gamma^\mu\boldsymbol{\pi}\psi \tag{2}$$

for torsion with

$$Y\nabla^2 R_{\mu\nu} + \tfrac{1}{2}(4Z+Y)\nabla^2 R g_{\mu\nu} - (2Z+Y)\nabla_\mu\nabla_\nu R + \\ + 2YR_{\mu\rho\nu\sigma}R^{\rho\sigma} - \tfrac{1}{2}YR_{\alpha\rho}R^{\alpha\rho}g_{\mu\nu} + 2ZRR_{\mu\nu} - \tfrac{1}{2}ZR^2 g_{\mu\nu} + R_{\mu\nu} - \tfrac{1}{2}g_{\mu\nu}R = \\ = \tfrac{1}{2}[\tfrac{1}{4}(\partial W)^2 g_{\mu\nu} - (\partial W)_{\nu\alpha}(\partial W)_\mu{}^\alpha + M^2(W_\mu W_\nu - \tfrac{1}{2}W^2 g_{\mu\nu}) + \\ + \tfrac{i}{4}(\overline{\psi}\gamma_\mu\boldsymbol{\nabla}_\nu\psi - \boldsymbol{\nabla}_\nu\overline{\psi}\gamma_\mu\psi + \overline{\psi}\gamma_\nu\boldsymbol{\nabla}_\mu\psi - \boldsymbol{\nabla}_\mu\overline{\psi}\gamma_\nu\psi) - \\ - \tfrac{1}{2}X(W_\nu\overline{\psi}\gamma_\mu\boldsymbol{\pi}\psi + W_\mu\overline{\psi}\gamma_\nu\boldsymbol{\pi}\psi)] \tag{3}$$

for gravity and

$$i\gamma^\mu\boldsymbol{\nabla}_\mu\psi - XW_\sigma\gamma^\sigma\boldsymbol{\pi}\psi - m\psi = 0 \tag{4}$$

for the spinor field. By taking the divergence of (2) and contracting (3) we get the constraints

$$M^2\nabla_\mu W^\mu = 2Xmi\overline{\psi}\boldsymbol{\pi}\psi \tag{5}$$

and

$$4(3Z+Y)\nabla^2 R - 2R = -M^2 W^2 + m\overline{\psi}\psi \tag{6}$$

which will be important in the following. Again, the case $Y = Z = 0$ reduces all to the least-order derivative form we would have in the usual Einsteinian second-order gravity, although in general the kinetic term ensures fourth-order character to gravitation. Because each curvature has the mass dimension equal 2, single-curvature kinetic terms of second-order gravity scale as $l^{-2}$ and double-curvature kinetic terms of fourth-order gravity scale as $l^{-4}$ with the consequence that Einstein gravity is more relevant in the case of large $l$ while its fourth-order extension is dominant for small $l$ in general. Consequently, the usual theory of gravitation is recovered in the infrared whereas some new physics is expected in the ultraviolet as we will see next.

*High Energy and Averaged Spins*

In the next pages we will be focusing on the problems usually met in the ultraviolet, that is at short distances, and hence high energies. In addition, since the situations we want to study (black holes and big bang) involve very large number of spinors with randomly distributed spin, an average on spin will also be assumed. When the high energy density condition is implemented, whenever short distances are considered, terms with high mass dimension become dominant as compared to all else, and thus such a condition of ultraviolet regimes can be implemented by requiring that only the mass dimension 4 terms are kept in all field equations. After doing this, we lose the mass and all single-curvature terms, remaining with

$$\nabla_\rho (\partial W)^{\rho\mu} = X \overline{\psi} \gamma^\mu \boldsymbol{\pi} \psi \tag{7}$$

for torsion and

$$\begin{aligned}
Y\nabla^2 R_{\mu\nu} + \tfrac{1}{2}(4Z+Y)\nabla^2 R g_{\mu\nu} - (2Z+Y)\nabla_\mu\nabla_\nu R + 2Y R_{\mu\rho\nu\sigma}R^{\rho\sigma} - \tfrac{1}{2}Y R_{\alpha\rho}R^{\alpha\rho}g_{\mu\nu} + \\
+ 2ZRR_{\mu\nu} - \tfrac{1}{2}ZR^2 g_{\mu\nu} = \tfrac{1}{2}[\tfrac{1}{4}(\partial W)^2 g_{\mu\nu} - (\partial W)_{\nu\alpha}(\partial W)_\mu{}^\alpha + \\
+ \tfrac{i}{4}(\overline{\psi}\gamma_\mu\nabla_\nu\psi - \nabla_\nu\overline{\psi}\gamma_\mu\psi + \overline{\psi}\gamma_\nu\nabla_\mu\psi - \nabla_\mu\overline{\psi}\gamma_\nu\psi) - \\
- \tfrac{1}{2}X(W_\nu\overline{\psi}\gamma_\mu\boldsymbol{\pi}\psi + W_\mu\overline{\psi}\gamma_\nu\boldsymbol{\pi}\psi)]
\end{aligned} \tag{8}$$

for gravity together with

$$i\gamma^\mu(\boldsymbol{\nabla}_\mu\psi + iXW_\mu\boldsymbol{\pi}\psi) = 0 \tag{9}$$

for the spinor field. The constraints are

$$i\overline{\psi}\boldsymbol{\pi}\psi = 0 \tag{10}$$

and

$$\nabla^2 R = 0 \tag{11}$$

as easy to see. Notice that condition $i\overline{\psi}\boldsymbol{\pi}\psi = 0$ is equivalent to the vanishing of the Yvon-Takabayashi angle, and this is expected in ultra-relativistic limits such as those obtained in massless conditions [18]. Condition $\nabla^2 R = 0$ will prove to be important later on. Averaging spins has the same effect of requiring that all instances of the spin axial-vector $\overline{\psi}\gamma_\mu\boldsymbol{\pi}\psi$ be neglected, which leaves

$$\nabla_\rho(\partial W)^{\rho\mu} = 0 \tag{12}$$

for torsion with

$$\begin{aligned}
Y\nabla^2 R_{\mu\nu} - (2Z+Y)\nabla_\mu\nabla_\nu R + 2Y R_{\mu\rho\nu\sigma}R^{\rho\sigma} - \tfrac{1}{2}Y R_{\alpha\rho}R^{\alpha\rho}g_{\mu\nu} + \\
+ 2ZRR_{\mu\nu} - \tfrac{1}{2}ZR^2 g_{\mu\nu} = \tfrac{1}{2}[\tfrac{1}{4}(\partial W)^2 g_{\mu\nu} - (\partial W)_{\nu\alpha}(\partial W)_\mu{}^\alpha]
\end{aligned} \tag{13}$$

for gravity and no more occurrences of spinors. This fact is understood by recalling that with no Yvon-Takabayashi angle nor spin axial-vector, we have that $P_\nu = m u_\nu$ [18], with the spinor contribution in the energy turning into a mass dimension 3 term that, again in high energy density conditions, will become negligible. These last equations, the last of which now implying the subsidiary condition

$$\nabla^2 R = 0 \tag{14}$$

in general, are the basis for the study of those symmetric situations that are at the foundation of modern cosmology, and that is the big bang and black holes. In detail, in the following we will consider two situations of symmetry given when the space-time is isotropic, a first of pure isotropy and a second where isotropy is accompanied by homogeneity. The first case, corresponding to stationary spherical symmetry, is the best suited to represent black holes (unless we want to consider the rotating case where axial symmetry should be used instead), and the second case, corresponding to maximal symmetry, is best suited to represent the scenarios of cosmological evolution.

## 3. Isotropic Spaces

The treatment of singularity formation revolves around ways to evaluate whether the Hawking-Penrose dominant energy condition $R_{\mu\nu} u^\mu u^\nu > 0$ is respected. For Einstein gravity the field equations are given in terms of the Ricci tensor $R_{\mu\nu}$ and so this condition can be examined rather straightforwardly, but in higher-order extensions the field equations have an altogether different form and therefore this direct study cannot generally be done. One alternative is that of focusing on a particular situation, described by a specific symmetry, finding exact solutions, and see that in such a case the dominant energy condition comes to be violated. This method is not general, as it is fully and strictly tied to the particular situation one wants to examine. Hence, one would not be able to prove that the dominant energy condition is always violated, but solely that it can be violated. Nevertheless, for now this is the most we can hope for. So, in the following, we shall focus on two particular (although well known) situations.

### 3.1. Black Holes

Let us begin the study by testing our equations against the case of black holes. This case will be considered not rotating, so to exploit spherical symmetry, and therefore, in polar coordinates, we have that the metric is given by

$$g_{tt} = A \quad g_{rr} = -B$$
$$g_{\theta\theta} = -r^2 \quad g_{\varphi\varphi} = -r^2 |\sin\theta|^2 \tag{15}$$

with $A(r)$ and $B(r)$ in general. The connection is

$$\Lambda^t_{tr} = \frac{A'}{2A} \quad \Lambda^r_{tt} = \frac{A'}{2B} \quad \Lambda^r_{rr} = \frac{B'}{2B}$$
$$\Lambda^r_{\theta\theta} = -\frac{r}{B} \quad \Lambda^r_{\varphi\varphi} = -\frac{r}{B}|\sin\theta|^2 \quad \Lambda^\theta_{\varphi\varphi} = -\cos\theta\sin\theta$$
$$\Lambda^\theta_{\theta r} = \Lambda^\varphi_{\varphi r} = \frac{1}{r} \quad \Lambda^\varphi_{\varphi\theta} = \cot\theta \tag{16}$$

from which covariant derivatives and curvatures can be computed. In particular, the curvatures are given by

$$R^{tr}{}_{tr} = \frac{A''}{2AB} - \frac{A'^2}{4A^2B} - \frac{A'B'}{4AB^2} \quad R^{t\theta}{}_{t\theta} = R^{t\varphi}{}_{t\varphi} = \frac{A'}{2ABr}$$
$$R^{r\theta}{}_{r\theta} = R^{r\varphi}{}_{r\varphi} = -\frac{B'}{2B^2r} \quad R^{\theta\varphi}{}_{\theta\varphi} = \frac{1}{Br^2} - \frac{1}{r^2} \tag{17}$$

with contraction

$$R^t{}_t = \frac{A''}{2AB} - \frac{A'^2}{4A^2B} - \frac{A'B'}{4AB^2} + \frac{A'}{ABr}$$
$$R^r{}_r = \frac{A''}{2AB} - \frac{A'^2}{4A^2B} - \frac{A'B'}{4AB^2} - \frac{B'}{B^2r}$$
$$R^\theta{}_\theta = \frac{A'}{2ABr} - \frac{B'}{2B^2r} + \frac{1}{Br^2} - \frac{1}{r^2}$$
$$R^\varphi{}_\varphi = \frac{A'}{2ABr} - \frac{B'}{2B^2r} + \frac{1}{Br^2} - \frac{1}{r^2} \qquad (18)$$

and contraction

$$R = \frac{A''}{AB} - \frac{A'^2}{2A^2B} - \frac{A'B'}{2AB^2} + \frac{2A'}{ABr} - \frac{2B'}{B^2r} + \frac{2}{Br^2} - \frac{2}{r^2} \qquad (19)$$

from which we calculate all terms in the field equations.

Torsion would have only $W_t(r)$ and $W_r(r)$ albeit only $W_t(r)$ can appear in the curl within the field equations.

However, we notice that precisely because of (12) there can be no dynamical solution for torsion, which can then be taken to vanish in the energy density contribution.

As another consequence of the field equations, (13) this time, we can pick $AB = 1$ without any loss of generality.

With this wisdom, we can proceed to observe that the constraint $\nabla^2 R = 0$ can be solved with some non-singular $R$ only if $R$ is a constant. To see this we have to consider that the Laplacian is generally expressed by

$$\nabla^2 R \equiv \frac{1}{\sqrt{|g|}} \partial_\mu \left( \sqrt{|g|} g^{\mu\nu} \partial_\nu R \right) \qquad (20)$$

where $|g|$ is the determinant of the metric, and so

$$\nabla^2 R = -\frac{1}{r^2} \left( r^2 B^{-1} R' \right)' \qquad (21)$$

for our metric and where the prime stands for derivative with respect to the radial coordinate. So $\nabla^2 R = 0$ can be integrated easily as

$$R' = aBr^{-2} \qquad (22)$$

with $a$ a constant. This yields a divergent behaviour for $R$ in the origin unless $B$ behaves as $r^b$ with $b > 2$ although in such a case the metric would degenerate in the origin and thus the solution would still be singular. Hope for a non-singular behaviour can only be found in having $a = 0$ and hence $R$ constant. Setting $R = 12k$ we get therefore

$$12k = A'' + 4A'/r + 2(A-1)/r^2 = [r^2(A-1)]''/r^2 \qquad (23)$$

which is now easy to solve. In fact

$$A = 1 + kr^2 \qquad (24)$$

is the only metric still non-singular. This metric is exact solution for the field Equation (13) in this particular case.

The above solution (24) in the limit $r \to 0$ becomes flat consequently showing no formation of singularity at all.

A point is worth commenting. The solution (24) represents what we expect to be the solution near the center of the matter distribution, that is black holes with spherical symmetry in conditions of high energy density, although this does not mean that (24) is the full solution. The full solution is a metric that has (24) as the limit behaviour at the center of the matter distribution where large energy densities are found. Because (24) is an exact

solution of the approximated field Equation (13), is a limiting case of the exact solution to the complete field Equation (3).

It is considerably difficult to obtain such a most general solution. But for the purposes we have here all we need is to assess its ultraviolet behaviour, that is for large energy density near the center of the material distribution.

*3.2. Big Bang*

Having worked out a rather general way to treat such a singularity problem in the case of black holes, we will move to study the case of the big bang. Now we have both isotropy and homogeneity, and so in polar coordinates

$$g_{tt} = 1$$
$$g_{rr} = -A^2 \quad g_{\theta\theta} = -A^2 r^2 \quad g_{\varphi\varphi} = -A^2 r^2 |\sin\theta|^2 \tag{25}$$

with $A(t)$ in general. The connection is

$$\Lambda^t_{rr} = A\dot{A} \quad \Lambda^t_{\theta\theta} = A\dot{A}r^2 \quad \Lambda^t_{\varphi\varphi} = A\dot{A}r^2 |\sin\theta|^2$$
$$\Lambda^r_{rt} = \Lambda^\theta_{\theta t} = \Lambda^\varphi_{\varphi t} = \frac{\dot{A}}{A}$$
$$\Lambda^r_{\theta\theta} = -r \quad \Lambda^r_{\varphi\varphi} = -r|\sin\theta|^2 \quad \Lambda^\theta_{\varphi\varphi} = -\cos\theta\sin\theta$$
$$\Lambda^\theta_{\theta r} = \Lambda^\varphi_{\varphi r} = \frac{1}{r} \quad \Lambda^\varphi_{\varphi\theta} = \cot\theta \tag{26}$$

from which covariant derivatives and curvatures can be computed. In particular, the curvatures are given by

$$R^{tr}{}_{tr} = R^{t\theta}{}_{t\theta} = R^{t\varphi}{}_{t\varphi} = -\frac{\ddot{A}}{A}$$
$$R^{r\theta}{}_{r\theta} = R^{r\varphi}{}_{r\varphi} = R^{\theta\varphi}{}_{\theta\varphi} = -\frac{\dot{A}^2}{A^2} \tag{27}$$

with contraction

$$R^t{}_t = -3\frac{\ddot{A}}{A}$$
$$R^r{}_r = R^\theta{}_\theta = R^\varphi{}_\varphi = -2\left(\frac{\ddot{A}}{2A} + \frac{\dot{A}^2}{A^2}\right) \tag{28}$$

and contraction

$$R = -6\left(\frac{\ddot{A}}{A} + \frac{\dot{A}^2}{A^2}\right) \tag{29}$$

from which we calculate all terms in the field equations.

Torsion has only $W_t(t)$ and therefore no contribution.

With the same reasoning as above we can try to search for the cases in which $R$ is constant. So setting $R = 12k$

$$-2k = \frac{\ddot{A}}{A} + \frac{\dot{A}^2}{A^2} = \frac{\ddot{A}^2}{2A^2} \tag{30}$$

or equivalently

$$\ddot{A}^2 + 4kA^2 = 0 \tag{31}$$

as a well known ordinary differential equation. With freedom to choose $k$ we may notice that for $k = -a^2$ we obtain

$$\ddot{A}^2 - 4a^2 A^2 = 0 \tag{32}$$

as the only field equation admitting non-vanishing solutions and with

$$A^2 = e^{2at} \tag{33}$$

as the only expanding solution. So

$$A = e^{at} \tag{34}$$

is the only possible solution that will remain non-singular at all times. Again, field Equation (13) are verified.

The above solution (34) in the limit $t \to 0$ becomes flat consequently showing no formation of singularity again, but now the specific type of solution has acquire an additional piece of information. In fact, allowing only solutions that do not degenerate the metric, that is excluding solutions of the type $\sin(at)$ or $\cos(at)$, not only we have eliminated the possibility of recurring universes, but we have also fixed the Ricci scalar to be negative. And this will have a very specific effect on the Ricci scalar as well.

In fact the Ricci tensor is $R_{\mu\nu} = -3a^2 g_{\mu\nu}$ and in turn $R_{\mu\nu} u^\mu u^\nu < 0$ for all curves of tangent vector $u^\alpha$ time-like.

This blunt violation of the Hawking-Penrose dominant energy condition is the clearest sign that the singularity formation is not longer a necessary feature of these types of extended theories of the gravitational field in general.

Finally, it may be worth mentioning that the solution (34) also gives rise to an era of inflationary expansion as the one we would expect to have in the early universe.

Notice that in this model such an evolution is obtained without any cosmological constant and this is important since any cosmological constant term has mass dimension zero and so it is negligible at high energy densities.

## 4. General Consideration

In the previous analysis, we have seen two cases where it was in fact possible to find exact solutions as metrics that were non-singular and asymptotically flat, such that the corresponding space-times violated the dominant energy condition needed for singularity formation. Hence, the formation of singularities was not a necessity in these circumstances. Being restricted to specific situations, the analysis was not general, and thus we could only show the possibility of a solution to the problem, and not its whole applicability. Just the same, the results are intriguing.

Apart from the symmetries of the specific situations at hand, the only other element playing a role was the existence of fourth-order differential gravitational field equations. The found solutions would in fact not be solutions in the least-order derivative Einsteinian gravitation. Nor would they work for higher-order derivative extensions based solely on the Ricci scalar. In fact, in this case the analogous of field Equation (13) would be given by

$$f'(R)R_{\mu\nu} - \tfrac{1}{2}f(R)g_{\mu\nu} = 0 \tag{35}$$

and hence

$$6a^2 f'(a^2) + f(a^2) = 0 \tag{36}$$

which is not generally verified. They can for special situations such as $f(R) = R^2$ but this would merely coincide with the case $Y = 0$ in the field equations used above.

The method we used is certainly tied to the underlying symmetries of the system, but it is a character of fourth-order theories of gravitation and *no other* one.

## 5. Conclusions

In this paper, we have recalled the fourth-order theory of gravitation showing, in high-energy average-spin configuration, that the field equations admit exact solutions of specific symmetries: we have studied cases of isotropy and isotropy and homogeneity.

Isotropy, that is spherical symmetry, was best suited to examine the behaviour we might find in black holes, for which we had demonstrated the existence of metrics that were non-singular near the origin of the radial coordinate, whereas isotropy and homogeneity, that is maximal symmetry, was best suited for examining the behaviour one might find at the big bang, for which we have proven the existence of metrics that are non-singular near the origin of time and whose structure provides the Ricci scalar with a negative value imposing the violation of the Hawking-Penrose energy conditions.

Such a procedure pertains only to fourth-order theories of gravitation and to no other theory in general cases.

However, the method is highly dependent on the symmetries of the system. Can this methodology be extended so to include more general situations?

**Funding:** This research received no external funding.

**Institutional Review Board Statement:** Not applicable.

**Informed Consent Statement:** Not applicable.

**Data Availability Statement:** Manuscript has no associated data in any repository.

**Conflicts of Interest:** The author declares no conflict of interest.

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
