# Peer review of "A Note on Singularity Avoidance in Fourth-Order Gravity"

_universe, doi:10.3390/universe8010051_

Round 1

Reviewer 1 Report

I have read this article carefully.  It is correct.  However, I have two significant comments:

  • The paper deals only with the gravity  classical model, while the term "renormalizability" is used only in quantum field theory.  Therefore, this term should be removed from the title of the paper and everywhere else where it occurs in the paper.  In the paper there is only the following result "in the proposed gravitational model there are no singularities". but this does not mean that the corresponding model in QFT will be renormalizable. A number of QFT models in planar space are non-renormalizable, but they do not have singularities in the transition to the ultraviolet limit.

  • Since the proposed model differs from General Relativity, and General Relativity is reliably experimentally verified for the investigated (low) energies, there is a natural question whether this gravitational model at low energies gives results close to General Relativity. This is Correspondence principle .

After appropriate corrections  the article can be published in the journal  "Universe".

Reviewer 2 Report

The author examines the issue of singularity avoidance in renormalizable fourth-order theories of gravity and shows the existence of non-singular black hole solutions and cosmological solutions in these theories. In both cases the author assumed particular symmetries, i.e. spherical symmetry in the case of black holes and isotropy and homogeneity for cosmological solutions. The obtained results, although restricted to the assumed symmetries are interesting. Before deciding about the suitability of this article for publication I have the following comments for the author:

i. In the high energy limit the gravitational field equations in (8) loose the GR contribution R_{\mu\nu} - (1/2)g_{\mu\nu}R. Can the author explain this? 

ii. Just before eq. (19) the author states that the constraint \nabla^2 R = 0 leads to a constant R for a non-singular R. How is this obtained? 

iii. The obtained solution A = 1 + kr^2 in eq. (20) is supposed to represent a black hole (without a singularity). Can the author explain why he considers this to be a black hole solution? 

Reviewer 3 Report

This note considers a particular renormalizable theory of gravity including torsion and higher derivatives, with the aim of finding solutions that avoid the formation of singularities. The search for solutions is carried out in a specific limit of “high energies and averaging over spins”, and further simplifying assumptions are made. Under these assumptions, the solutions found are just maximally symmetric spaces (Anti de Sitter, Minkowski or de Sitter, depending on the value of k in Section 3.1, and just de Sitter in Section 3.2), though they are not recognized as such. These spaces are  non-singular in an obvious way.
The assumptions made are so constraining that the solutions found are trivial, hence the conclusions are essentially contentless. I therefore cannot recommend publication in Universe.

Round 2

Reviewer 2 Report

The author explains how the GR contributions are lost at high energies, and his explanation makes sense. However looking at the Lagrangian in (1), one notes that the Einstein-Hilbert term should be multiplied by some dimensional constant with units l^{-2}. The author also explains how one gets R=constant from \nabla^2 R = 0, and again the explanation makes sense. However I do not agree with his claim that the solution in (24) has to be regarded as a limiting black hole solution to eq. (13) which is only valid in the limit r->0. This is due to the fact that the condition R=constant which leads to (24) is supposed to be valid everywhere (in the high energy limit). In other words the author has not be able to establish the existence of a non-singular black hole in this theory, and since singularity avoidance was supposed to be the main result of the paper, I will not be able to recommend this paper for publication. 

Reviewer 3 Report

The Author is asking for a clarification of why in my report I claim the solutions found are trivial. My point is that under the assumptions made, that include the decoupling of the spinor dynamics and triviality of the torsion, it is immediate to see that any Einstein space solves Eq. (13). The solution given in Sections 3.1 and 3.2 are maximally symmetric spaces (de Sitter, Minkowski, or Anti de Sitter), which are the simplest instances of non-singular Einstein spaces. 
In my opinion, the analysis is too limited to provide original scientific progress, which is one of the requirements for publication in Universe. In particular, I do not see how this provides a way to attack the issue of singularity avoidance from a general perspective, as the Author claims in the abstract. 
